# Immunosuppressive niche engineering at the onset of human colorectal cancer

Chandler D. Gatenbee [1✉], Ann-Marie Baker [2], Ryan O. Schenck [1,3], Maximilian Strobl [1], Jeffrey West [1], Margarida P. Neves[2], Sara Yakub Hasan[2], Eszter Lakatos [2], Pierre Martinez[2,4], William C. H. Cross[2], Marnix Jansen [5], Manuel Rodriguez-Justo [5], Christopher J. Whelan [6,7], Andrea Sottoriva [8], Simon Leedham[3], Mark Robertson-Tessi [1,9], Trevor A. Graham [2,9] & Alexander R. A. Anderson [1,9✉]

The evolutionary dynamics of tumor initiation remain undetermined, and the interplay between neoplastic cells and the immune system is hypothesized to be critical in transformation. Colorectal cancer (CRC) presents a unique opportunity to study the transition to malignancy as pre-cancers (adenomas) and early-stage cancers are frequently resected. Here, we examine tumor-immune eco-evolutionary dynamics from pre-cancer to carcinoma using a computational model, ecological analysis of digital pathology data, and neoantigen prediction in 62 patient samples. Modeling predicted recruitment of immunosuppressive cells would be the most common driver of transformation. As predicted, ecological analysis reveals that progressed adenomas co-localized with immunosuppressive cells and cytokines, while benign adenomas co-localized with a mixed immune response. Carcinomas converge to a common immune "cold" ecology, relaxing selection against immunogenicity and high neoantigen burdens, with little evidence for PD-L1 overexpression driving tumor initiation. These findings suggest re-engineering the immunosuppressive niche may prove an effective immunotherapy in CRC.

[1] Integrated Mathematical Oncology Department, H. Lee Moffitt Cancer Center & Research Institute, 12902 Magnolia Drive, SRB 4, Tampa, FL 336122, USA. [2] Evolution and Cancer Laboratory, Centre for Genomics and Computational Biology, Barts Cancer Institute, Queen Mary University of London, London EC1M 6BQ, UK. [3] Wellcome Centre for Human Genetics, University of Oxford, Oxford OX37BN, UK. [4] Lyon Cancer Institute, Lyon, France. [5] Department of Pathology, University College London Hospital, London, UK. [6] Cancer Physiology, H. Lee Moffitt Cancer Center & Research Institute, 12902 Magnolia Drive, SRB 4, Tampa, FL 336122, USA. [7] Department of Biological Sciences, University of Illinois at Chicago, 845 West Taylor Street, Chicago, IL 60607, USA. [8] Center for Evolution and Cancer, Institute of Cancer Research, London, UK. [9] These authors jointly supervised this work: Mark Robertson-Tessi, Trevor A. Graham, Alexander R. A. Anderson. ✉email: chandler.gatenbee@moffitt.org; Alexander.Anderson@moffitt.org

The classical model of colorectal carcinogenesis is the adenoma-carcinoma pathway that describes the accumulation of (epi)mutations in benign (non-invasive) adenomas, which underpin the development of invasive carcinoma[1,2]. However, while the risk of developing colorectal cancer (CRC) is certainly increased by adenoma formation[3,4], it appears that few adenomas actually progress to cancer in a human lifetime: bowel cancer screening programs detect approximately five "high-risk" adenomas for every cancer found[4,5], and longitudinal endoscopic surveillance of adenomas reveals that <2% of adenomas progress to cancer within 3 years[6]. Consequently, there appears to be a substantial "evolutionary hurdle" that must be overcome for an adenoma to become invasive.

Immune predation is known to modulate and indeed suppress neoplastic growth[7]. As in lung cancer[8,9], the evolution of immune evasion is therefore likely to be a key barrier on the evolutionary path to CRC. Newly arising somatic mutations in a tumor may generate neoantigens, which can then serve as targets for immune-cell recognition and destruction (in particular CD8+ cytotoxic T lymphocytes). However, the negative selective pressure imposed by the immune system provides positive selection pressure for strategies to avoid elimination, a process known as immunoediting[10,11]. Such immune evasion has been described as a hallmark of cancer[12], and there are many mechanisms by which tumor cells may escape immune predation including, but not limited to, blockade of cytotoxic T cell attack via expression of programmed death-ligand 1 (PD-L1), recruitment of immunosuppressive cells such as macrophages and neutrophils, and disruption of the antigen presentation machinery[13–17].

In CRC, multiple lines of evidence suggest a critical role for immunological surveillance in regulating tumor growth. The density of tumor-infiltrating T cells is highly prognostic, with greater infiltration associated with a better prognosis[18,19]. Moreover, non-metastatic CRC has an increased level of T cell infiltration as compared to metastatic CRC[20]. Genomic analysis reveals that a higher predicted neoantigen burden is associated with increased tumor lymphocyte infiltration[9,21,22]. Large-scale genomic analysis indicates immune evasion mechanisms have evolved in the majority of CRC[11]. Immune modulation studies in mouse models of CRC also provide support for a critical regulatory role of the immune system in colorectal carcinogenesis[23–25].

The progression from colorectal adenoma (CRA) to CRC likely requires the accumulation of multiple (epi)genetic aberrations. The overall single-nucleotide alteration (SNA) burden appears comparable between CRA and CRC, including SNAs for putative driver genes, with the exception of TP53[26]. Moreover, analysis of the evolutionary dynamics of sub-clones within CRC indicates a frequent lack of differential selection operating between subclones[27], suggesting that at the point of invasion, the founder cancer cell had already acquired all the alterations necessary for its malignant phenotype and also that the bulk of tumor cells was not experiencing markedly differential immune predation. Thus, malignant potential and immune evasion may be established together when cancer growth is initiated, and conceivably immune evasion could be the key phenotypic trait governing the transition from adenomas to cancers.

By integrating mathematical modeling, ecological analysis of whole slide images, and multi-region neoantigen prediction, here we have investigated the role of immune escape in the evolution of CRC from precursor CRA. We hypothesized that immune surveillance represents a key hurdle that prevents the outgrowth of invasive cells within a benign CRA. For the neoplastic cells, the acquisition of mutations responsible for progression must be balanced against the risk of accumulating too many neoantigens that would lead to immune elimination. To investigate this idea, we developed a mathematical model that simulates tumor evolution under immune predation and escapes in order to define the expected patterns of immune activity and antigenic intratumor heterogeneity (aITH) throughout tumor progression from benign to malignant. We then looked for these signatures in a cross-sectional cohort of CRA as well as early- and later-stage CRC using the expression of 17 markers (by immunohistochemistry (IHC) and RNA in situ hybridization (ISH)) and called neoantigens from multi-region whole-exome sequencing (WES) data. We leveraged several ecological tools to describe and compare the cellular composition of tumors as biological units, providing a holistic view of how tumors change through progression. Comparison of the model and data indicates a key role for immune evasion at the onset of malignancy in CRC.

## Results

**Immune suppression is the superior escape strategy.** Ecologists have long used mathematical models to simulate the dynamics of interacting species. Lotka-Volterra models, in particular, have been used to study predator–prey dynamics, competition, mutualism, and amensalism[28–30]. Such models have frequently been adopted by mathematical oncologists to study the evolution of resistance under different therapeutic regimens, as well as tumor–immune interactions[31–34]. Here we combine various forms of deterministic Lotka-Volterra models to understand the role of the immune system in tumor initiation and progression under immune predation and subsequent escape from immune control via two distinct "strategies." The first strategy, Blockade, gives tumor cells the ability to effectively neutralize cytotoxic T cells by blocking their attack. Two biological examples of Blockade would be PD-L1 and PI-9, the first of which inhibits cytotoxic T cells[35], while the latter inhibits cytotoxic T cell-induced apoptosis by blocking the perforin/granzyme B pathway[36]. The second strategy, Suppression, gives tumor cells the ability to recruit immunosuppressive cells, such as M2 macrophages. Many of these immunosuppressive cells are also involved in wound repair and thus create a microenvironment that not only suppresses the anti-tumor immune response but also promotes cellular growth via angiogenesis, production of growth factors (e.g., epidermal growth factor), and matrix metalloproteinases[37–39].

In our Lotka–Volterra model of competition, predation, mutualism, and amensalism, tumor cells compete with one another, are preyed upon by cytotoxic T cells, and are supported by "mutualist" immunosuppressive cells that suppress immune attack and promote growth:

$$\frac{dN_i}{dt} = N_i \left[ r_i \left( \frac{K_i - \overbrace{\sum_1^j \alpha_{ij} N_j}^{\text{competition}} + \overbrace{0.5 \sigma_i N_i}^{\text{growth benefit}}}{K_i} \right) - \gamma_i \overbrace{\left(1 - \frac{\sigma_i}{\gamma_i}\right)}^{\text{suppression}} \overbrace{\left(1 - \phi_i\right)}^{\text{blockade}} - \delta \right]$$

(1)

Each $i$th subpopulation of tumor cells, with population size $N_i$, is composed of cells that have four unique traits:

(1) Antigenicity ($\gamma_i$), which defines the immune kill rate and is determined by the collection of neoantigens carried by the cells in the $i$th subpopulation[40];

(2) Degree of Blockade ($\phi_i$), a cell-intrinsic mechanism that reduces the effective killing rate by cytotoxic T cells on the $i$th subpopulation;

(3) Degree of Suppression ($\sigma_i$), which determines the ability of tumor cells to recruit immunosuppressive cells; this has a dual effect of reducing immune kill and enhancing growth for the $i^{th}$ population (reviewed in refs. [13,14,16,17]);

(4) Species ($j$), determined by the number of driver mutations accumulated by each cell in the stochastic model. Populations

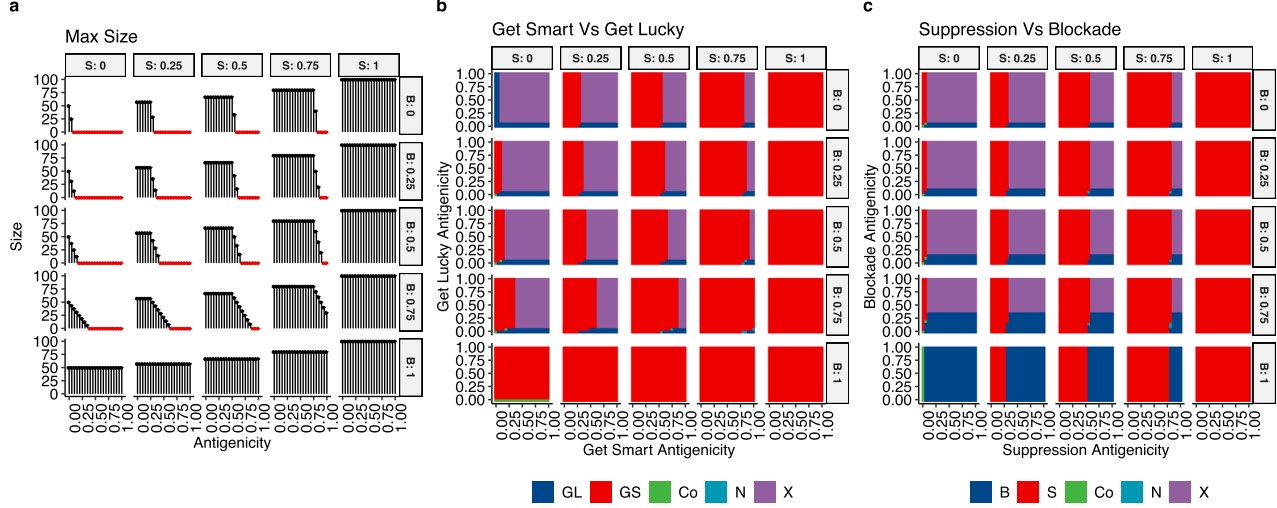

**Fig. 1 Model results predict that immune suppression is the dominant escape mechanism. a** Maximum size (i.e., number of cells) of each strategy for different values of suppression (S, $\sigma$, columns), blockade (B, $\phi$, rows), and antigenicity (inset $x$-axes). Red dots indicate that the population is eliminated by immune predation, highlighting that clones must have some way to avoid immune predation, either by having low antigenicity (Get Lucky) or actively mitigating attack (Get Smart). As population growth was simulated using an epithelial division rate, these results indicate immune escape should be an early event if the clone is to survive increased predation due to the accumulation of neoantigens associated with the mutation. **b, c** Outcome of competition, where "Co" means neither strategy would go extinct and could co-exist, "N", indicates either strategy could win, but the one that does must have a larger starting population size, and "X" means that neither population survived immune predation. **b** Outcomes of competition between Get Lucky (GL) and Get Smart (GS), over a range of suppression values (S, $\sigma$), blockade values (B, $\phi$), and antigenicities ($\gamma$), such that for GL $\sigma = 0$, $\phi = 0$, and for GS $0 \leq \phi \leq 1$, $0 \leq \phi \leq 1$. Get Smart wins out over Get Lucky 71% of the time (7778 out of 11025 parameter combinations). **c** Outcomes of competition between Suppression (S) and Blockade (B), over a range of suppression values (S, $\sigma$), blockade values (B, $\phi$), and antigenicities ($\gamma$), such that for P $\sigma = 0$, $0 \leq \phi \leq 1$, and for S $\phi = 0$, $0 \leq \sigma \leq 1$. Out of the 11,025 parameter combinations, Suppression (S) wins the majority of the time, 55% (6047 parameter combinations), with Blockade (B) winning 17% of the time (1862 parameter combinations). Combined, these results highlight that Get Smart almost always wins out over Get Lucky, while Suppression wins against Blockade. This indicates that having an immune escape strategy (Get Smart), particularly immune suppression, significantly increases a clone's fitness, allowing it to sweep through the population, often by engineering an immunosuppressive niche.

with <2 driver mutations are "normal"; those with 2 or 3 mutations are adenomas (CRA), while those with 4+ mutations are carcinomas (CRC). The species of the population determines the population's division rate, carrying capacity, and interactions with other tumor populations (see next section for more details).

Each subpopulation has a distinct carrying capacity ($K_i$, the maximum viable size of the subpopulation), division rate ($r_i$), and interactions with other species, defined in the interaction matrix, **α**. We assume that all subpopulations have the same intrinsic death rate, $\delta$.

In simulations of the model, two distinct trajectories emerged. The first trajectory, which we dub "Get Lucky", is when tumor cells acquired only mutations that had low antigenicities but lacked an active escape mechanism ($\sigma_i = 0, \phi_i = 0$) (Fig. 1a, top left panel). The second trajectory, termed "Get Smart", is when tumors acquired one or more mutations that facilitated escape from elimination by the immune system ($\sigma_i > 0$ and/or $\phi_i > 0$); this, in turn, reduced the selection pressure against high antigenicity mutations (Fig. 1c). These two trajectories demonstrated that it was necessary for tumor cells to mitigate immune attacks if they were to grow. Cells that did not avoid immune predation (i.e., having antigenicity too high and having no/weak escape mechanisms) were always eliminated (red dots in Fig. 1a).

Tumors are heterogeneous, and it is likely that cells following the Get Smart and Get Lucky trajectories would, at some point, co-exist in a single tumor. To determine the outcome of Get Lucky versus Get Smart competition, which would define the tumor's susceptibility to immune attack (and thus immunotherapies), we conducted a steady-state analysis of the pairwise competition between the Get Lucky strategy ($\sigma_i = 0, \phi_i = 0$), and the range of possible Get Smart strategies (combinations of $\sigma_i \geq 0$,

$\phi_i \geq 0$) (Fig. 1b). This competition was conducted over a range of antigenicities ($0 \leq \gamma_i \leq 1$) for each strategy, which allowed us to examine the role of the immune response in determining which strategy would win and which would be eliminated. We assumed that both strategies existed within the same "species" of tumor cell, and thus had the same division rate, shared a carrying capacity, and, aside from the strategy, neither population had any intrinsic advantage over the other (i.e., $\alpha_{ij} = 1$).

The Get Lucky versus Get Smart analysis indicated that out of the 11,025 parameter combinations, in 71% of the simulations Get Smart out-competed Get Lucky, regardless of the starting sizes of each population. Get Lucky only won 3.3% of the time, which occurred only when the competing Get Smart population's protection was insufficient to overcome its elevated antigenicity. Given that coexistence was rare (0.25% of simulations), these results suggest that clones which have the ability to escape immune attack (Get Smart) will sweep through the population, resulting in the extinction of immune-susceptible clones.

Given that Get Smart is expected to almost always outcompete Get Lucky, we next determined under which inflammatory conditions each individual escape strategy, Blockade or Suppression, could (co)exist. Suppression populations were modeled by setting $0 \leq \sigma_i \leq 1$ and keeping $\phi_i = 0$, while Blockade populations were modeled by setting $0 \leq \phi_i \leq 1$ and keeping $\sigma_i = 0$. By conducting steady-state analyses over all combinations of $\sigma_i, \phi_i, \gamma_i$ (each varying between 0 and 1 in increments of 0.05), we could thus determine the outcome of all pairwise competition models of Blockade versus Suppression, each subject to their own immune pressures as determined by their antigenicity.

The Blockade versus Suppression analysis revealed that immune suppression was the superior strategy, as it outcompeted

Blockade in the majority of simulations (55%, 6047 out of 11025 simulations) due to the ability to both increase growth and reduce immune kill rates, which resulted in a higher net growth rate (Fig. 1d, e and Supplemental Fig. 1). Only rarely did initial population size matter (0.054% of simulations, Fig. 1c), as Suppression won frequently even when it had a smaller starting size than Blockade. As in the Get Lucky versus Get Smart analysis, coexistence between the individual strategies was rare (0.25%). Together, these observations suggest that clones which can recruit immunosuppressive cells should sweep through the population, even outcompeting cells capable of blocking immune attack directly.

These modeling results suggest that active immune escape mechanisms (as opposed to passive escape by getting lucky) should frequently be observed in established cancers and that the recruitment of immunosuppressive cells that also increase tumor cell viability is expected to be common among successful tumors.

**Immune suppression expected to occur early, increase risk, and shape intra-tumor antigenic heterogeneity.** While deterministic Lotka-Volterra models are readily tractable, they are limited in that they simulate interactions between a fixed number of species, meaning that they cannot simulate the evolution of novel clones, nor generation of intra-tumor heterogeneity. In order to simulate the emergence and subsequent evolution of tumor clones, we created an evolutionary branching process version of the model, which allows us to determine the timing of immune escape, how much it increases the risk of progression to cancer, and how the ecology and aITH changes through the progression of CRC. We also simulate the accrual of somatic mutations (see supplemental methods), the evolution through "tissue compartments" (from healthy tissue (E) into an adenoma (CRA) and then carcinoma (CRC)), and explicitly represent the interactions with immune cells (Fig. 2a).

Simulations are initiated with a homogenous, non-antigenic epithelial population that lacks immune-escape mechanisms. Each cell acquires new antigenic mutations at a rate $\mu$ per division, and strict inheritance of mutations means a new mutant daughter cell (and its subsequent clone) will always be more antigenic than its parent since it carries all parental mutations along with the new mutation. Cells can also acquire the ability to block the immune attack ($\phi_i > 0$), recruit immunosuppressive cells ($\sigma_i > 0$), or gain a single driver mutation at per-division rates of 3.7e−6 for blockade and suppression, and 9.26e−5 for gaining a driver gene (Fig. 2b, and see Supplemental materials for details on the calculation of mutation rates). We assumed an epistatic model of progression[41], such that the acquisition of two driver mutations defined the transition from E to CRA, and the acquisition of a further two driver mutations defined the transition to CRC. We modeled competition among species based on the space they occupied: E and CRA clones have no interactions because adenomas grow superficially to the epithelium, whereas we assumed that CRC subpopulations were strongly interacting with both E and CRA populations to describe an overgrowth of the adenoma and invasion through the epithelium (Fig. 2a).

We assume that the number of immunosuppressive cells will be proportional to the product of the immunosuppressive cell recruitment parameter and the size of the immunosuppressive cell-recruiting population ($N_i \sigma_i$). Likewise, we assume that the number of cytotoxic T cells reactive to a subpopulation would be proportional to the subpopulation size, its antigenicity, and ability to recruit immunosuppressive cells, $N_i \gamma_i \left(1 - \frac{\sigma_i}{\gamma_i}\right)$. We can therefore calculate the number of immunosuppressive cells, cytotoxic T cells, and the number of cells with the immune

blockade. Figure 2c shows a representative simulation run of the model.

In order to understand the role of immune blockade and suppression in determining the risk of cancer development, we conducted a parameter sweep over the ranges of blockade strength ($\phi$) and suppression strength ($\sigma$). This also allowed us to see how each strategy affects aITH and also when they emerge in the timeline of cancer progression. The sweeps ranged from 0 to 1 by increments of 0.04 for each of the two parameters. A mutation rate of 2.91e−9 was used in accordance with measurements from Werner et al.[42] We furthermore conducted 100 runs of each parameter set to be able to explore the effects of the stochasticity in the model.

There were four possible outcomes of each simulation: (1) The tissue remained healthy for 100 years, which occurred when either no CRA/CRC populations evolved, or remained under 100 cells in size if they did evolve; (2) a CRA evolved but was eliminated; (3) a CRC evolved but was eliminated; (4) a CRC evolved from a CRA, and the CRC existed for 1 year, thus considered malignant. In Cases 2 and 3, the simulations were concluded after these elimination events, and the time was recorded.

As we observed in the previous model (Fig. 1), the Get Smart strategy dominated the CRC outcome in this expanded model. We examined the immune escape status of these populations in more detail, using a total 21,723 simulations where a CRC formed. Recall there are 3 Get Smart strategies (immunosuppression, blockade, or immunosuppression + blockade), and in 75.87% of those simulations, the ability to suppress immune attack was an early event; immunosuppression was a trait of the precursor CRA, meaning that the first CRC cell arising from a driver mutation already possessed the immunosuppressive phenotype (Fig. 3a). When the precursor CRA relied on a Get Lucky strategy, the emerging CRC was founded by a clone with the ability to suppress immune attack in 8.72% of simulations. Across all parameter combinations, only 1.45% of CRC (and thus their pre-cursor CRA) were successfully established without active immune escape (i.e., these tumors had completely followed the "Get Lucky" trajectory).

Greater immune suppression, which resulted from stronger recruitment of anti-inflammatory pro-growth immune cells, significantly increased the risk and rate of progression from CRA to CRC (Fig. 3b, c). In contrast, no matter its strength, the ability to reduce immune kill through increased blockade had little effect on risk and rate of progression. Tumors also became more antigenic as immune suppressive mechanisms became more effective. This phenomenon occurs because immune suppression relaxes selection against high antigenicity, allowing more antigenic mutations to persist (Fig. 3d).

**Immunosuppressive niche construction begins early.** We next examined primary human tumors to determine the dominant escape trajectory in CRC, and the time at which it emerged, using model predictions to guide the analysis. Specifically, we considered 21 CRAs, 15 CRCs, and 26 "carcinoma-in-adenoma" (CIA) samples, the latter of which consist of a nascent carcinoma (C-CIA) adjacent to an adenoma (A-CIA) (Figs. 4–7). As such, we consider the A-CIA samples as progressed/late adenomas, as they are likely either the precursor of the adjacent carcinoma or in a similar state as that carcinoma's actual precursor.

We performed multi-color IHC analysis to characterize the immune microenvironment (tumor ecology) in all $n = 86$ cases (62 samples, with each of the 26 CIA samples having both an A-CIA and C-CIA region) (Fig. 4 and Supplemental Figs. 2–6). We used multi-region WES to measure neoantigen intra-tumor

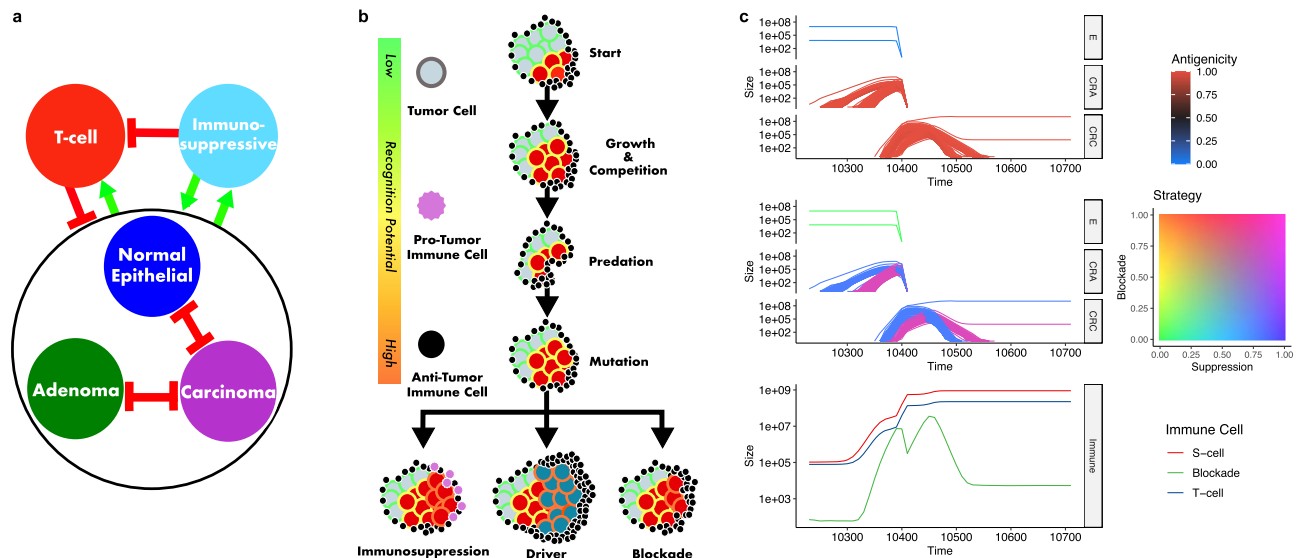

**Fig. 2 The competition model in Eq. (1) was expanded to include mutation, allowing for simulation of evolution from pre-tumor epithelium (E) to adenoma (CRA) to carcinoma (CRC), with the possibility to acquire driver mutations and immune escape strategies (blockade or suppression). a** This model simulates cell–cell competition, predation by cytotoxic T cells, mutualism between tumor cells and immunosuppressive cells, and reduction of T cell infiltration by immunosuppressive cells (amensalism). Red bars indicate inhibitory interactions while green arrows represent positive interactions. **b** The model is initiated with a large population of homogenous non-immunogenic epithelial cells. Mutation may occur during division, resulting in the creation of a new population that inherits all ancestral antigens and mutations from its parent. Each mutation is accompanied by the generation of a neoantigen, which stimulates increased attack by cytotoxic T cells at a rate proportional to the neoantigen's randomly assigned recognition potential. With low probability, cells may also acquire one of three beneficial mutations: (1) the ability to recruit immunosuppressive cells, which decreases T cell attack while increasing growth rates; (2) the ability to block T cell attack, reducing immune kill; and (3) acquisition of driver mutations, which increase division rates and carrying capacities once enough have accumulated. If an epithelial population acquires two driver mutations it is considered an adenoma, which grows in a separate niche atop the epithelial tissue, limiting interactions between the two populations. CRA become carcinomas when they acquire four driver mutations in total. CRC has the ability to grow both atop and into the epithelium, allowing carcinomas to invade and destroy epithelial and adenoma clones. Due to inheritance, there may eventually be clones that accumulate multiple beneficial mutations. **c** Example simulation, where an E population evolved into a CRA after 1390 days, which in turn evolved into a CRC. The dominant CRC phenotype had a high antigenicity (top) and strong immunosuppression (middle), resulting in an increase in immunosuppressive cells and a drop in T cells (bottom). Over 200,000 phenotypes developed in this simulation, but only those populations with >50,000 cells are shown.

heterogeneity (aITH) in a subset of $n = 16$ cases (160 exomes, as each was downsampled 10 times to normalize depth, for $n = 6$ CRA, $n = 3$ CIA, and 7 CRC).

Our modeling predicts that early acquisition of immune suppression is the dominant escape strategy in CRC, and is crucial for progression. Given these expectations, we selected a panel of 17 markers to describe the immune response in each tumor type (Fig. 4, Supplemental Figs. 2 and 3). Samples were stained in two sets: the first set comprised 12 CRA, 25 CIA, and 15 CRC, with IHC staining for tumor cells (CK), cytotoxic T cells (CD8), neutrophils (elastase), macrophages (CD68), B cells (CD20), immune checkpoint inhibition (PD-L1), vasculature (CD31), proliferation (Ki67), fibroblasts (αSMA), DNA damage (γH2AX), and COX2. The second set of samples comprised 9 CRA, 9 CIA, and 9 CRC, with IHC staining for macrophages (CD68), M1 macrophages (iNOS) and M2 macrophages (CD163), and RNA ISH for expression of immunosuppressive cytokines IL-10 and TGF-β, and inflammatory cytokines TNF-α and IL-6. Nine CIA and 9 CRC were in both sets and therefore were stained with all 17 markers. These cases are referred to as the "Intersection Set".

We then examined how cell-type abundances, and the tumor and microenvironment as a whole, changed through progression. The Virtual Alignment of pathoLogy Image Series (VALIS) registration software[43] was used to create whole-slide composite images from the multi-color IHC and/or RNA ISH (Fig. 4, Supplemental Figs. 2 and 3). This composite image was then divided into 250 μm × 250 μm quadrats, and the number of pixels positive for each stain was determined for each quadrat (Fig. 4

and Supplemental Figs. 4–6). Quadrat counts were used to perform a spatial analysis, while the abundance of each cell type was approximated by dividing the total number of positive pixels by the total area sampled. See Supplemental Figs. 7 and 10 for details of the statistical tests and significance values.

Our modeling predicted an increase in immunosuppressive cells and a drop in anti-tumor inflammation during the evolution from A-CIA to C-CIA to CRC. Correspondingly, the immunosuppressive cytokine IL-10 showed a significant increase in expression through progression, and this correlated with greater numbers of putatively tumor-promoting neutrophils (elastase) and macrophages (CD68) (Fig. 5). Anti-tumor cytotoxic CD8 T cells were progressively excluded through progression (Fig. 5a). There was also a significant drop in TNF-α through progression, a powerful pro-inflammatory cytokine[44]. These patterns were directly evident within individual CIA lesions (e.g., comparing A-CIA directly to its descendant C-CIA). These changes culminate in an immunosuppressed, pro-growth ecology that supports a higher abundance of tumor cells (CK) in CRC, having significantly less inflammation (TNF-α and CD8) and more pro-tumor factors, including macrophages (CD68), TGF-β, and vasculature (CD31) as compared to benign adenomas (CRA). Together, these data suggest that CRA elicits a strong anti-tumor immune response, and the evolution of an immunosuppressive microenvironment is associated with progression to CRC.

We next used ecological methods to describe and compare the overall cellular compositions of tumors as a whole, as opposed to one cell type/cytokine at a time. Indicator species analysis (ISA)

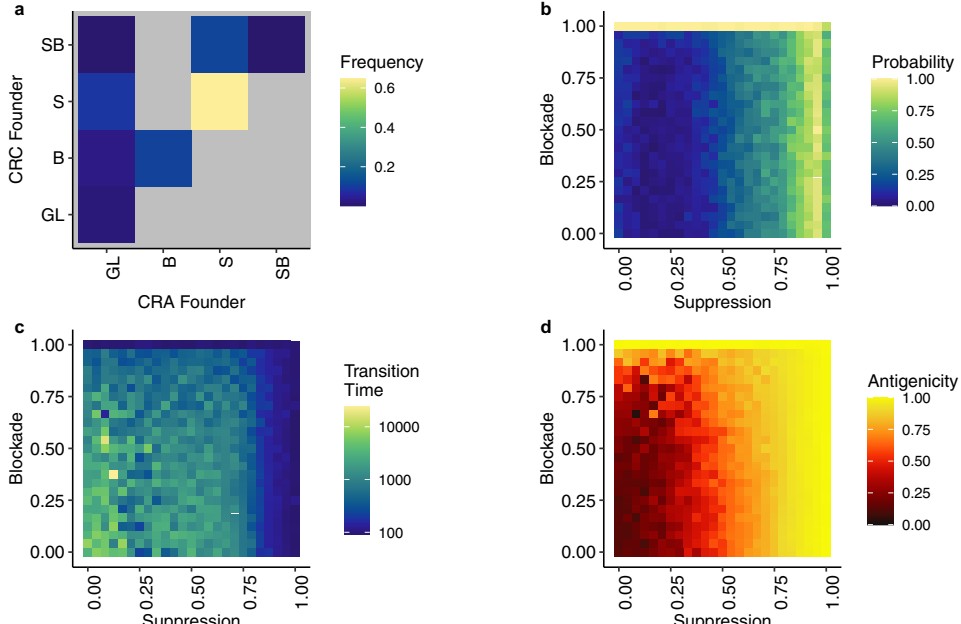

**Fig. 3 Immune escape occurs early and increases the risk of tumor formation. a** Comparison of the adenoma (CRA) founder's immune escape phenotype to the eventual founding branch of the carcinoma (CRC) population, where GL = founder relied on Get Lucky, B = founder had Blockade, S = founder had Suppression, and SB = founder had Suppression and Blockade. Out of the 21,723 simulations, 75.87% of the simulations had the acquisition of Suppression as an early event, occurring in the CRA founder and later inherited by the CRC founder. **b** The probability of a carcinoma forming under various strengths of immune escape. Probability is calculated as the number of times a carcinoma existed for 1 year, divided by the number of times that parameter set was run, in this case, 100 times each. The gradient from left to right indicates that immune suppression increases the probability of tumor formation, while lack of a gradient from top to bottom suggests that immune blockade has little effect on the probability. **c** The average amount of time, in days, between CRA and CRC formation, i.e., the transition time, for each parameter set. Tumors that progressed most rapidly from CRA to CRC are those with an immunosuppressive strategy, which reduces immune predation and relaxes selection against high antigenicity. **d** Averages of each malignant tumor's mean antigenicity (weighted by each population's size) for each parameter combination. The horizontal gradient indicates that immune suppression allows for increased antigenicity, as immune suppression relaxes selection against immunogenicity.

determines which cell types (or factors), if any, uniquely define a group of tumors[45]. First, we explored the cell types that defined tumor stages. ISA revealed that a defining characteristic of benign CRA is an abundance of CD8+ T cells and PD-L1 expression, consistent with the notion that CRA remains under immune attack but avoids elimination due to having a protective mechanism (Fig. 6a). Indicator species characteristic of CRC were an abundance of tumor cells, increased vasculature, and potentially-immunosuppressive neutrophils. Alternative analysis using constrained principal coordinates (CAP) and permutational multivariate analysis of variance (PERMANOVA)[46–48] (Fig. 6c) confirmed these results. This further reiterates that CRA is defined by cytotoxic T cells, while CRC is defined by immune suppression.

If tumors are engineering an immunosuppressive niche to escape immune predation, as predicted by the model, we would expect to see a correlation between differences in tumor microenvironment and differences in immune composition. We tested for this relationship using the Mantel test[49] to detect correlations between the ecological dissimilarities in immune-cell lineages (T cells, B cells, macrophages, and neutrophils) and dissimilarities in non-immune cells of the microenvironment (tumor cells, fibroblasts, PD-L1, and vasculature) that could affect the behavior of the aforementioned immune cells. Results indicate that differences in immune composition correlate significantly with differences in non-immune ecology ($p = 0.003$), indicating there is a significant relationship between environmental similarity and immunological similarity (Supplemental Fig. 8).

Our initial modeling (Fig. 1) predicted that immune suppression is under strong positive selection, as it offers a significant advantage over other escape strategies. This selective process

would be expected to decrease both intra-tumor and intra-stage heterogeneity, as we would expect a shift from a mixed immune response (pro- and anti-inflammatory cells) to one dominated by immunosuppressive cells (lower intra-tumor heterogeneity), resulting in a cold immune ecology common to carcinomas (low intra-stage heterogeneity). To test these predictions, we used Simpson's index to measure intra-tumor heterogeneity, which was then compared across stages, and found a significant drop in ecological diversity from CRA to CRC ($p = 0.0052$, Jonckheere-Terpstra test for decreasing trends) (Fig. 6b).

Intra-stage heterogeneity was calculated using PERMDISP2[50], which tests for significant differences in intra-group homogeneity across groups, where here groups are tumor stages. This analysis revealed that CRCs are more homogeneous than precursor A-CIA and C-CIA (Fig. 6d). That is, CRC has ecologies more similar to one another than A-CIA are to one another or C-CIA are to one another. This signal of low intra-CRC heterogeneity is consistent with the hypothesis that CRC has converged to a shared, immune cold ecology.

Like CRC, CRA intra-stage heterogeneity is also lower than that of A-CIA and C-CIA. This finding is surprising as, consistent with modeling, one would expect that had they not been resected, some of the observed CRA would progress while others would remain benign, due to having different immune ecologies. We believe the lower intra-stage heterogeneity of CRA may arise for several reasons. First, while CRA has similar mixes of cells (resulting in low intra-stage heterogeneity), it may be that the more important factor is the spatial configuration of cells within the tumor (Fig. 7). Second, progression from adenoma to carcinoma is rare[4,5], and so the signal of any outliers (i.e., CRA

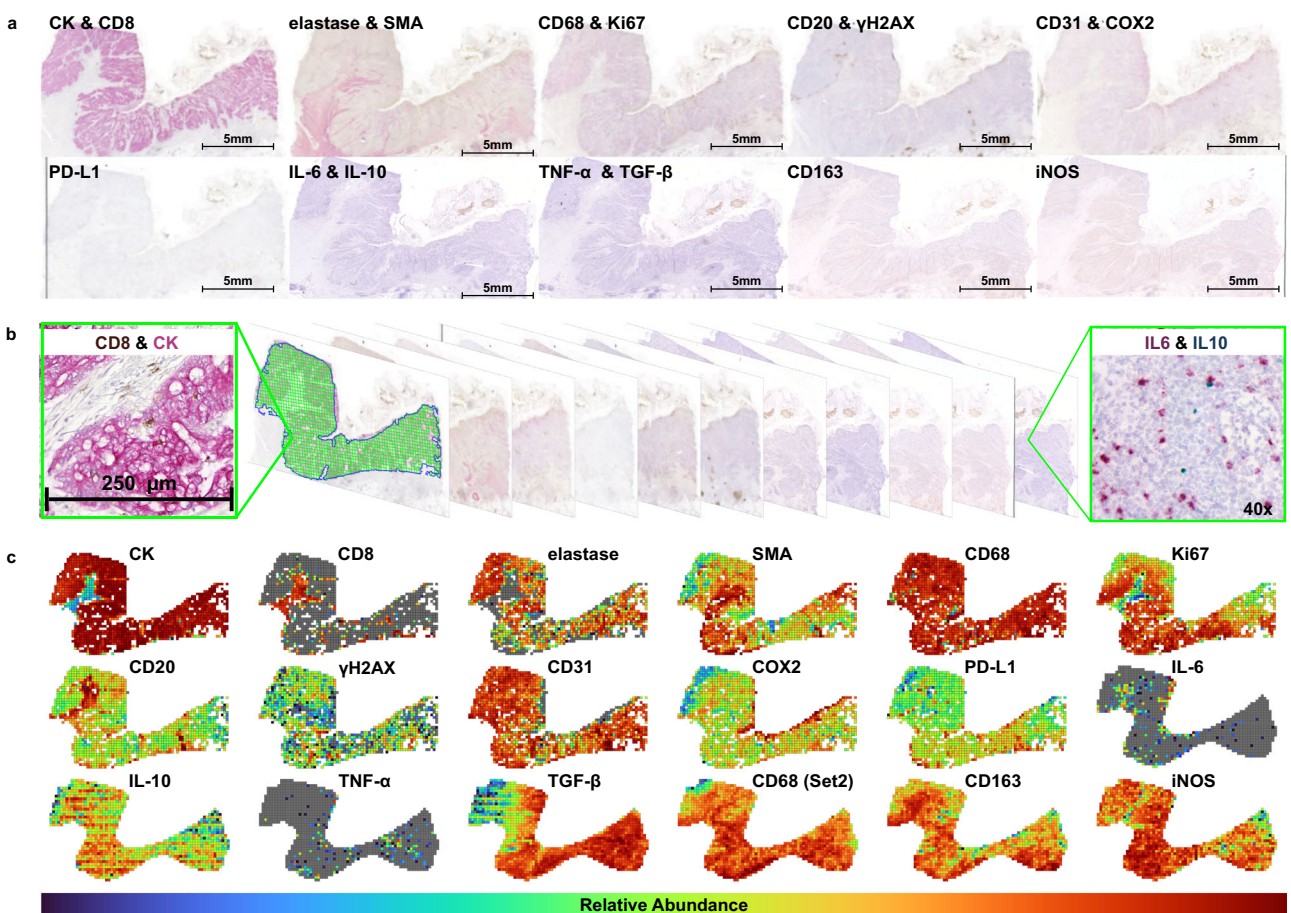

**Fig. 4 Example of a single sample used to describe the tumor ecology. a** To describe the spatial distribution of cell types and gene expression, 10 serial slices were taken from each sample and each stained with one or two markers. There were two sets of serial slices, one with CK, CD8, elastase, SMA, CD68, Ki67, CD20, γH2AX, CD31, COX2, and PD-L1. The second set included IL-6, IL-10, TNF-α, TGF-β, CD163, CD68, and iNOS. **b** After aligning the images in the two sets, each slice was divided into quadrats of 250 μm × 250 μm, and stain segmentation was performed to determine the abundance of each marker at ×40 magnification. **c** Quadrat counts for all 18 markers in one colorectal carcinoma (CRC) sample (there are 17 unique markers, but CD68 is in both Set 1 and Set 2). The same process was repeated for $n = 12$ colorectal adenomas (CRA), $n = 26$, "carcinoma-in-adenoma" (CIA), and $n = 15$ CRC, using the first set of markers, and for $n = 9$ CRA, $n = 9$ CIA, and $n = 9$ CRC using the second set of markers. This process was repeated for all samples, allowing us to describe changes in the tumor ecology, cell abundance/gene expression, and spatial associations. Values in each plot are scaled to reflect the minimum and maximum values of that marker in the image. See Supplemental Fig. 3 for more detailed images of each marker and Supplemental Figs. 4–6 for figures showing quadrat counts for all samples.

that would have progressed) is low, meaning that most CRA has similar ecologies. Third, modeling predicts that the transition to CRC is rapid when immune suppression is strong (Fig. 3c), and so the chances of catching CRA at the very earliest stages of transition could be rare. As is the case with the second hypothesis, the signal from such rare CRA would be drowned out by the majority of CRA that remain under immune control.

We believe the higher intra-stage heterogeneity of A-CIA and C-CIA is related to their being resected at different points in the transition to an immune cold ecology. As many microenvironmental changes would occur during tumor evolution and construction of the immunosuppressive niche, it would be expected that the resected CIA samples (A-CIA, C-CIA) are at different points along their transition to an immune cold ecology, and thus exhibit the observed high intra-stage heterogeneity. However, all of the microenvironmental changes eventually lead to the cold immune ecology shared by CRC, hence a lower intra-stage heterogeneity found in the final CRC stage.

**Carcinomas are isolated from immune attack**. We next determined how spatial associations between cell types and cytokines

change during tumor progression (Fig. 7c–f, Supplemental Figs. 9 and 10). We created "species association networks"[51] that quantify the spatial co-localization between cell types using the data derived from quadrat counts (Fig. 7). This method has an advantage over some other spatial statistics, such as correlation or nearest-neighbor distances, as each network describes the conditional dependence relationships between all species, rather than only pairwise spatial distances or correlations in isolation. By accounting for the spatial distribution of all cell types at once, these networks are able to remove indirect effects (i.e., cases where species A and B are correlated because they both interact with C but do not interact with each other) that can confound some other spatial statistics. By controlling for these indirect effects, we have more confidence that the spatial associations are valid. The result is a description of how species are associated in space, after controlling for all other species in the dataset. The resulting complex collection of individual pairwise spatial associations (Supplemental Fig. 9) was then organized into meaningful cell type/cytokine spatial clusters by performing community detection on each stage's averaged network (Fig. 7c–f). This was accomplished using the Leiden algorithm to detect communities on each tumor stage's multiplex graph (one

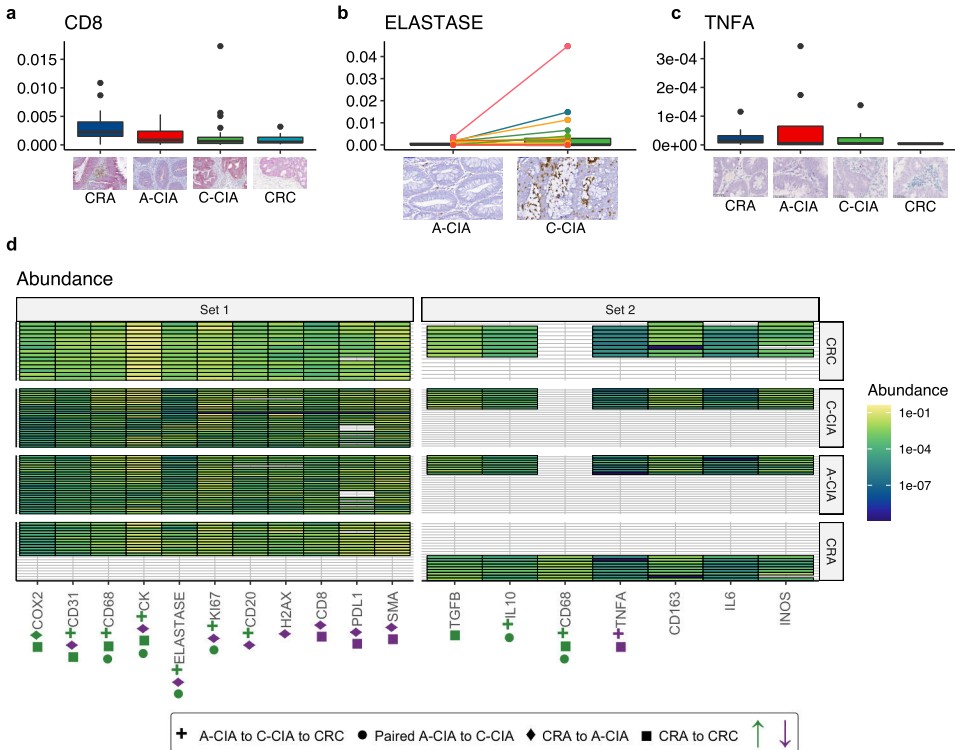

**Fig. 5 Significant changes in average cell type abundance and cytokine expression profiles show that immunosuppressive niche engineering is observed early at the progressed adenoma (A-CIA) stage and continues through colorectal carcinoma (CRC).** In **a–c**, the center line of each boxplot indicates the median, the top and bottom of the box indicate the 75th and 25th percentiles, respectively, the top whisker the largest value that is no further than 1.5 interquartile range (IQR) from the 75th percentile, the bottom whisker the smallest value no more than 1.5 IQR from the 25th percentile, and points indicate outliers. In addition to showing the distribution of abundances, we also provide histological examples of differences in abundance/gene expression in each category. Results shown in **a**, **b** and Set 1 in **d** are based on n = 12 independent colorectal adenomas (CRA), n = 25 independent A-CIA, n = 26 independent early carcinomas (C-CIA), and n = 15 independent CRC. Results in **c** and Set 2 in **d** are based on comparing n = 9 independent CRA, n = 8 independent A-CIA, n = 9 independent C-CIA, and n = 9 independent CRC. **a** Benign adenomas (CRA) have significantly higher cytotoxic T cell abundances than A-CIA or carcinomas (CRC). **b** Changes that accompany the transition from adenoma to carcinoma were revealed using paired tests to directly compare adenomas (A-CIA) to their descendant carcinomas (C-CIA). Here, we show the significant increase in neutrophils from each A-CIA to its descendant C-CIA. **c** Expression of the inflammatory cytokine TNF-α (red) significantly decreased during progression from progressed adenoma (A-CIA) to early carcinoma (C-CIA) to malignancy (CRC), culminating in CRC having significantly less TNF-α, and more immunosuppressive TGF-β (blue) than benign adenomas (CRA). **d** Table showing the abundance of each marker (columns) for each sample (rows). Shapes below each marker signify that there were significant changes in abundances across the specified group. Green indicates that the marker increased from the first group to the last group, while purple indicates that the abundances decreased. These results are based on a suite of statistical tests. Please see Supplemental Fig. 7 for p values.

Examination of changes in cell and cytokine abundances reveal that, among other things, there is a significant decrease in the abundance of cytotoxic T cells from benign adenoma (CRA) to adenomas that progressed (A-CIA) (purple diamond). Paired tests reveal that when directly compared to their ancestral adenoma (A-CIA), carcinomas (C-CIA) have significant increases in the abundance of macrophages (CD68), neutrophils (elastase), and cytokine IL-10, all of which can be associated with immune suppression. These trends are also observed when quantifying changes from late adenoma (A-CIA) to nascent carcinoma (C-CIA) to mature carcinoma (CRC), along with additional increases in B cells (CD20) and vasculature (CD31), and accompanied by a decrease in the inflammatory cytokine TNF-α. Together, these patterns indicate that during the evolution of a malignant tumor, the engineering of an immunosuppressive ecology begins early in the adenoma stage, and continues through to malignant CRC formation.

graph of average positive associations and the second graph of average negative associations)[52,53].

Benign adenomas (CRA) showed signs of being locked in a battle between pro- versus anti-inflammatory cells, as tumor cells clustered with cytotoxic T cells and macrophages, which in turn clustered with both anti-tumor M1 (iNOS) and pro-tumor M2 (CD163) macrophage markers (cluster 3-4 in Fig. 7c). It is worth noting that macrophages are plastic and can express both M1- and M2-associated markers, so the observation that the CRA macrophages express both markers suggests that they were not polarized to the extremes of either the M1 or M2 phenotypes[54]. CRA was the only stage where tumors clustered with cytotoxic T cells, and also had significantly higher abundances of cytotoxic T cells (compared to A-CIA and CRC), and inflammatory TNF-α (compared to CRC) (Fig. 5). Together, these observations suggest

that the immune system is successfully mounting an attack against tumor cells (cytotoxic T cells, M1 macrophages), despite tumor cell PD-L1 expression and presence of M2 macrophages (CD68 and CD163).

In contrast to benign adenomas (CRA), the spatial organization of progressed adenomas (A-CIA) was broadly distinct, nevertheless with some similar individual pairwise spatial associations (Supplemental Figs. 9 and 10). Instead of clustering with cytotoxic T cells, tumor cells in A-CIA cluster with macrophages (CD68) and B cells (CD20), both of which can produce IL-10 (also found in the A-CIA tumor cluster). IL-10 is known to polarize macrophages towards the M2 phenotype (cluster 3 in Fig. 7d)[38,55–57]. The macrophages in A-CIA also cluster with the immunosuppressive cytokine TGF-β, which can too polarize macrophages towards the M2 phenotype. These

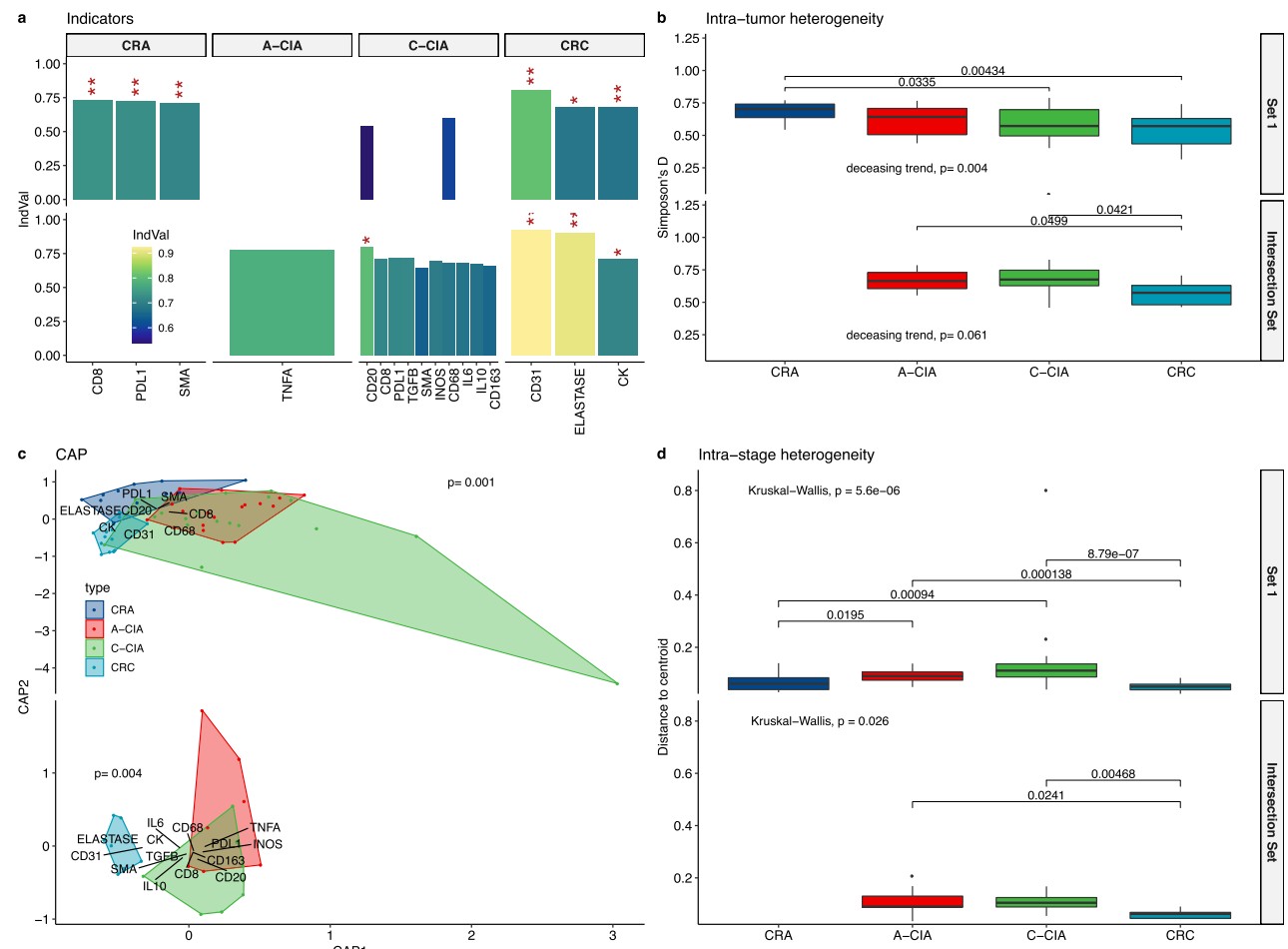

**Fig. 6 Whole-tissue ecological analysis reveals adenomas and carcinomas have unique and distinct hot and cold ecologies, respectively.** Set 1 included $n = 12$ independent colorectal adenomas (CRA), $n = 26$ "carcinoma-in-adenoma" (CIA = A-CIA & C-CIA), and $n = 15$ independent colorectal carcinomas (CRC). The Intersection Set included $n = 9$ CIA and $n = 9$ CRC. **a** Indicator species analysis was performed to determine which, if any, cells or cytokines define each tumor stage. The indicator value (IndVal) describes how strongly each cell/cytokine defines the tumor stage. CRA is uniquely defined by cytotoxic T cells (CD8), PD-L1, and fibroblasts (SMA), while CRC is defined by tumor cells (CK), vasculature (CD31), and neutrophils (elastase). **b** Intra-tumor heterogeneity, as measured by Simpson's index. Intra-tumor heterogeneity decreases, as might be expected if immune suppression reduces the abundance of pro-inflammatory cells (Jonckheere–Terpstra test for decreasing trends). **c** Plotting results from constrained analysis of principal coordinates (CAP) of ecological dissimilarities reveals that CRA and CRC have unique mixes of cell types that create significantly distinct ecologies, as determined by PERMANOVA tests. **d** As determined by the two-sided Kruskal–Wallis rank-sum test, CRC ecologies are significantly more similar to one another than the precursor A-CIA and C-CIA, suggesting that CRC have converged to a common immune-cold ecology defined by vasculature, tumor cells, and neutrophils. This was determined using PERMDISP2, which describes the amount of intra-stage ecological heterogeneity by comparing intra-group dispersions of ecological dissimilarities. In the boxplots in **b**, **d**, the center line indicates the median, the top and bottom of the box indicate the 75th and 25th percentiles, respectively, the top whisker the largest value that is no further than 1.5 Interquartile range (IQR) from the 75th percentile, the bottom whisker the smallest value no more than 1.5 IQR from the 25th percentile, and points indicate outliers. Pairwise significance of differences was determined with Dunn's test of multiple comparisons using rank sums.

observations strongly suggest these macrophages are of the immunosuppressive M2 type (cluster 4)[38]. With the exception of TNF-α, clusters 3 & 4 suggest that tumor cells in progressed adenomas exist within a predominantly immunosuppressive niche, with low CD8 T cell infiltration (note the negative spatial association between CD8 and CK).

Tumor cells in nascent carcinomas (C-CIA) appear to be highly proliferative, clustering with Ki67 (cluster 1, Fig. 7e), while having weak spatial associations with immune cells (clusters 2), a trend that continues in CRC. Compared to benign adenomas, tumor cells in CRC have significantly weaker spatial associations with cytotoxic T cells (Supplemental Fig. 9), and significantly lower abundances of cytotoxic T cells and TNF-α (Fig. 5), suggesting that the anti-tumor response has been suppressed in

CRC, which frees tumor cells to divide at higher rates (increased CK/Ki67 spatial association).

**Benign adenomas are highly immunogenic, while immune cold carcinomas exhibit antigenic neutrality.** As neoantigens elicit an immune response, it should be expected that changes in the tumor ecology that alter the immune composition will lead to changes in aITH. Our modeling suggests that under immune predation the average neoantigen recognition potential should be low, as tumors must rely on the "Get Lucky" strategy. However, immune escape reduces the selection pressure against high antigenicity, permitting the existence of clones bearing neoantigens that have high recognition potentials (Figs. 1a and 3d). With the

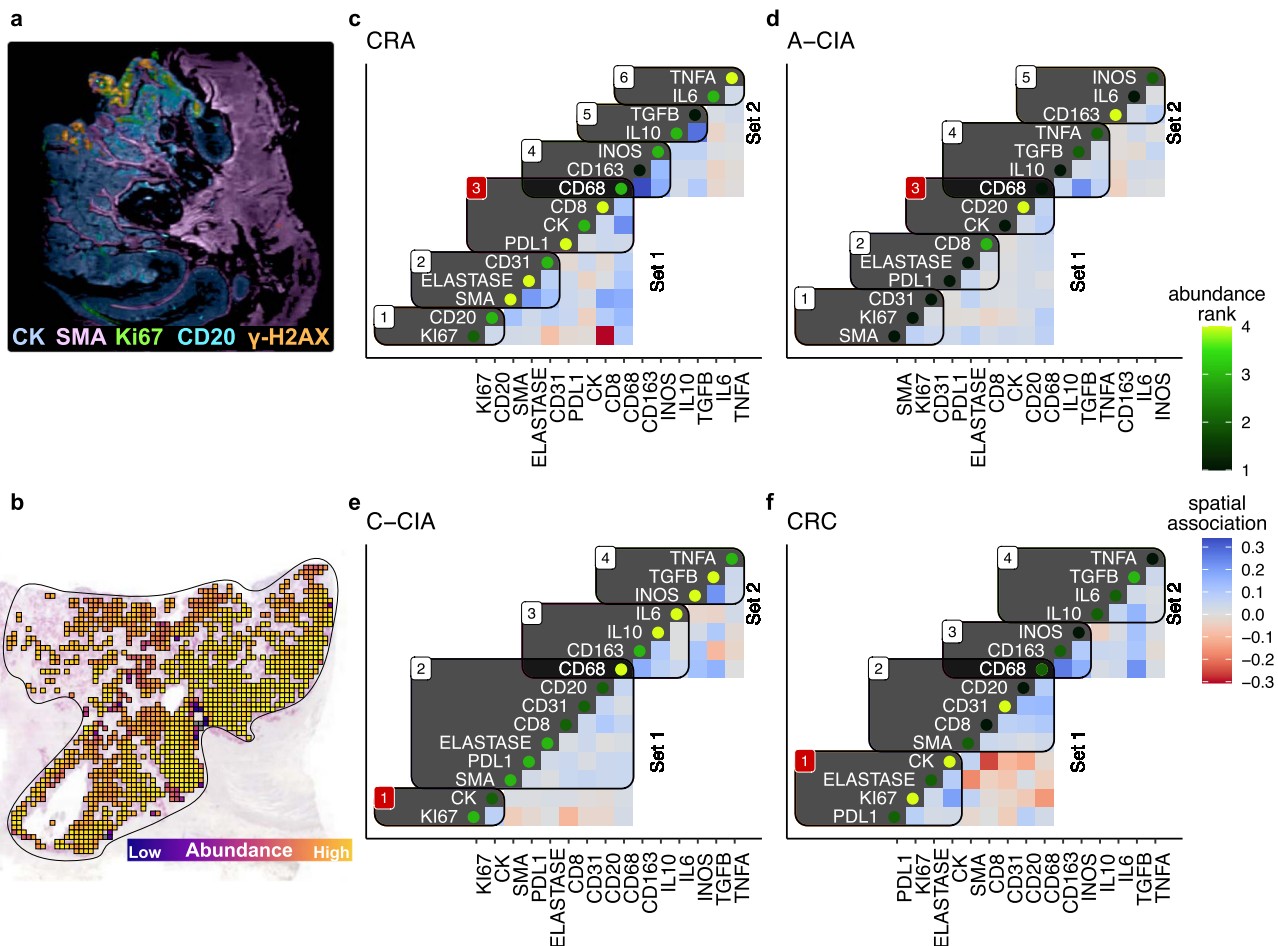

**Fig. 7 Changes in direct spatial associations. a** Composite images were created by aligning each sample's collection of serial slices. This example shows only five markers, but all markers were used for the spatial analysis. **b** Each composite image was then divided into quadrats 250 microns in width and height, wherein each marker was quantified at ×40 magnification. This image shows the quadrat positions overlaid on one image in the aligned series. **c–f)** Averaged spatial association networks, clustered by spatial communities (black boxes), with the tumor community labeled in red. The Set 1 samples included $n = 10$ colorectal adenomas (CRA), $n = 16$ progressed adenomas (A-CIA), $n = 17$ early carcinomas (C-CIA), and $n = 13$ colorectal carcinomas (CRC), while the Set 2 samples included $n = 8$ CRA, $n = 6$ A-CIA, $n = 9$ C-CIA, and $n = 7$ CRC. The order of cell types within each community does not have meaning. The color of each element in the lower diagonal indicates the average spatial association between the marker pair, with positive blue values indicating clustering, and negative red values meaning cell types/cytokines are found in separated areas. The color of the circle on the diagonal denotes the ranked abundance, e.g., CD8 is highest in CRA (yellow circle) and lowest in CRC (black circle). The interaction networks for both sets have been overlaid on one another using CD68, the only marker found in both datasets (Set 1 and Set 2). Tumor cells in CRA cluster with a mix of pro- and anti-tumor immune cells, suggesting that an anti-tumor immune response is being mounted despite the presence of immune escape mechanisms. In contrast, tumor cells in progressed adenomas (A-CIA) reside in a predominantly immunosuppressive niche, being associated with B cells (which can promote an M2 macrophage phenotype) and macrophages, which are in turn clustered with immunosuppressive cytokines (IL-10, TGF-β).

relaxation of selection pressures, antigenicity effectively becomes a quasi-neutral trait, resulting in an increase in aITH and neoantigen burden. Given this understanding of the relationship between the tumor ecology and aITH, we next used multi-region neoantigen prediction from WES to qualitatively infer the selection pressures experienced during each stage of progression.

CRA was the most immunogenic of all samples, with significantly higher mean neoantigen recognition potentials, when weighting each recognition potential by its associated neoantigen's variant allele frequency (VAF) (Fig. 8a). Conversely, CRC had the highest number of predicted neoantigens of any tumor stage (Fig. 8b), but fewer neoantigens at high VAF which also had high recognition potential. These observations are consistent with our modeling predictions that neoantigens are free to accumulate following escape from immune predation, and our ecological measurements indicate the establishment of a T cell-excluded immunosuppressive microenvironment in CRC, but not adenomas.

We used principal component analysis (PCA) to gain a holistic picture of the tumor-immune eco-evolutionary dynamics by examining aITH, tumor ecology, and tumor stage together (analysis restricted to samples for which we had both genomic and histological data; Fig. 8c). The first two components, which explained 67.3% of the variance (44.5 by PC1, and 22.8% by PC2), were positively correlated, with negative PC1 and PC2 associated with signs of a cytotoxic T cell response associated with high weighted recognition potentials. Larger PC1 and PC2 values were associated with growth and immune suppression (proliferation, vasculature, macrophages, neutrophils) and an increased neoantigen burden that would be expected to accompany a loss of selection against antigenicity, much like how neutral mutations accumulate freely[58]. Benign CRA and malignant CRC samples prove to be opposites, with CRA clustering in the inflammatory "corner," and CRC clustering in the opposite immunosuppression, pro-growth corner.

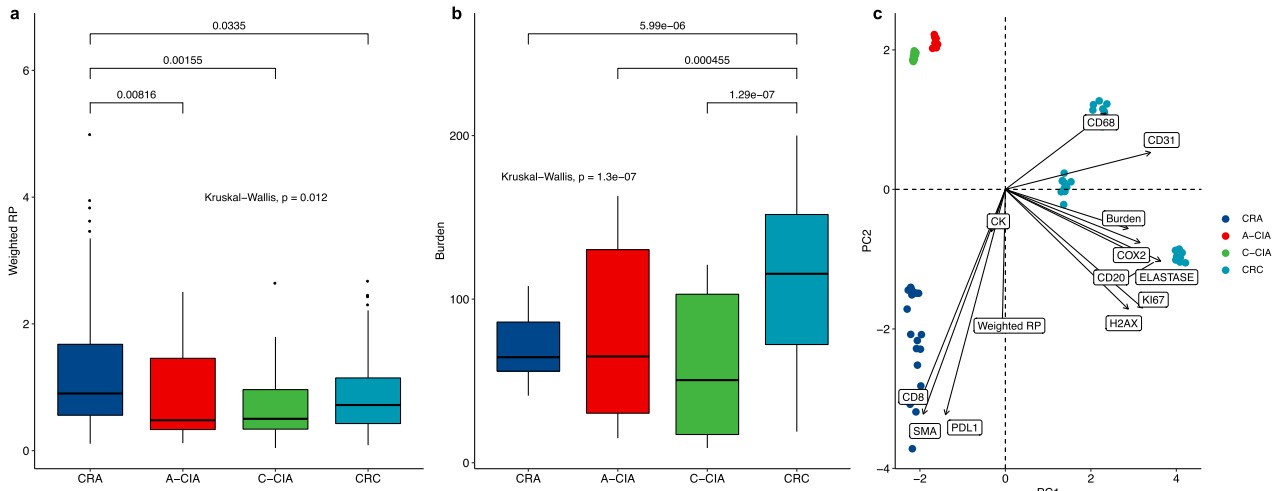

**Fig. 8 Observed patterns of aITH indicate benign adenomas (CRA) are highly antigenic and remain under immune control, while colorectal carcinomas (CRC) have escaped elimination via immune suppression.** Sample sizes are $n = 6$ CRA, $n = 3$ "carcinoma-in-ad" (CIA = A-CIA and C-CIA), and $n = 7$ CRC. In the boxplots shown in **a**, **b**, the center line of each boxplot indicates the median, the top and bottom of the box indicate the 75th and 25th percentiles, respectively, the top whisker the largest value that is no further than 1.5 Interquartile range (IQR) from the 75th percentile, the bottom whisker the smallest value no more than 1.5 IQR from the 25th percentile, and points indicate outliers. **a** Distribution of neoantigen recognition potentials (RP), weighted by their variant allele frequency (VAF). Comparisons across groups indicate that currently benign CRA are significantly more antigenic than all other tumor types, as determined using Dunn's test of multiple comparisons using rank sums. **b** CRC have significantly higher neoantigen burdens than all other cell types, and modeling indicates that this occurs when selection against antigenicity is relaxed due to immune suppression. **c** For the samples in which there was both imaging and genomics ($n = 2$ CRA, $n = 1$ CIA, $n = 3$ CRA, each with 10 downsampled replicates, thereby normalizing for sequencing depth), principal component analysis (PCA) of all cell/environmental markers, neoantigen burden, and weighed recognition potentials reveals that CRA is highly antigenic and inflammatory, while CRC is associated with immunosuppressive cells and high neoantigen burden.

**Model predicts benign adenomas are unable to overcome immune attack despite signs of immune suppression.** A surprising finding of the ecological analysis was that progressed (A-CIA) and benign adenomas (CRA) exhibit similar levels of some immunosuppressive markers, such as similar abundances of macrophages (CD68), M2 markers (CD163), and immunosuppressive cytokines (IL-10, TGF-β), with CRA even having significantly stronger positive spatial associations between CD163/CD68 (M2 macrophages) (Fig. 5, Supplemental Figs. 9 and 10). However, this appears to be counterbalanced by an anti-tumor inflammatory response, with CRA (compared to A-CIA) having significantly higher levels of CD8 and weighted neoantigen recognition potentials, whilst also forming spatial clusters consisting of a mix of pro- and anti-inflammatory cells (Figs. 5–8).

Both progressed and benign adenomas (A-CIA and CRA, respectively) show signs of immune suppression. We hypothesized that the key enabler of progression to invasive disease is the development of more effective immune suppression by eventually invasive lesions that is sufficient to overcome the higher immunogenicity of the progressed lesions. To test the plausibility of this hypothesis, we returned to our model to compare benign and progressed adenomas in simulations where benign adenomas had similar/higher levels of immune suppression and greater cytotoxic T cell infiltration and immunogenicity compared to progressed adenomas. We used a Monte Carlo accept-reject method to find model parameters that reproduced our observations that, compared to progressed adenomas (A-CIA), benign adenomas (CRA) were: significantly more immunogenic (higher CD8 abundances, higher neoantigen recognition potentials, residing in spatial niche containing CD8 cells and M1 macrophages); had more immune blockade (PD-L1); similar levels of immunosuppressive cells/cytokines (CD68, CD163, IL-10, TGF-β) (data analyses in Figs. 5–8, Supplemental Figs. 9 and 11).

The parameters that reproduced our observations all had a narrow range of high immune suppression values, 0.58–0.83

(25th and 75th percentiles), but a wide range of immune blockade values 0.17–0.71 (25th and 75th percentiles) (Fig. 9a). Within these simulations, the difference between adenomas that progressed and those that remained benign was that the former had a higher ratio of immune suppression to antigenicity (Fig. 9b–d). Going back to the data, we then quantified the ratio of immunosuppressive cytokines (TGF-β, IL-10) to inflammatory cytokines (TNF-α) in benign and progressed adenomas (Fig. 9e, f). In both cases, we found that progressed adenomas had significantly higher ratios of immunosuppressive cytokines to inflammatory cytokines. These findings are consistent with the model prediction that a key difference between progressed and benign adenomas is that immune suppression is more effective in progressed adenomas due to them having a reduced inflammatory response, making it easier to overcome. This builds upon the model's general conclusion that immune suppression is expected to be present at tumorigenesis, adding that immune suppression must also be sufficiently strong to overcome the tumor's immunogenicity, which is most easily accomplished by having immune suppression coupled with low immunogenicity: the pattern observed in progressed adenomas (A-CIA).

Additionally, model fitting also suggests that immune suppression is fairly strong in CRC, as most fitted values had high levels of immune suppression (Fig. 9a). It seems also that the role of PD-L1 is minimal, as there was a wide range of immune blockade strengths that fit the data. In other words, high protection from blockade occurred as frequently as low protection, likely because it is playing a secondary, almost supportive, role to immune suppression, and therefore the added benefit is minimal.

## Discussion

Integration of mathematical modeling, ecological analysis of whole slide images, and quantification of intra-tumor antigenic heterogeneity (aITH) paint a clear picture of the origins and

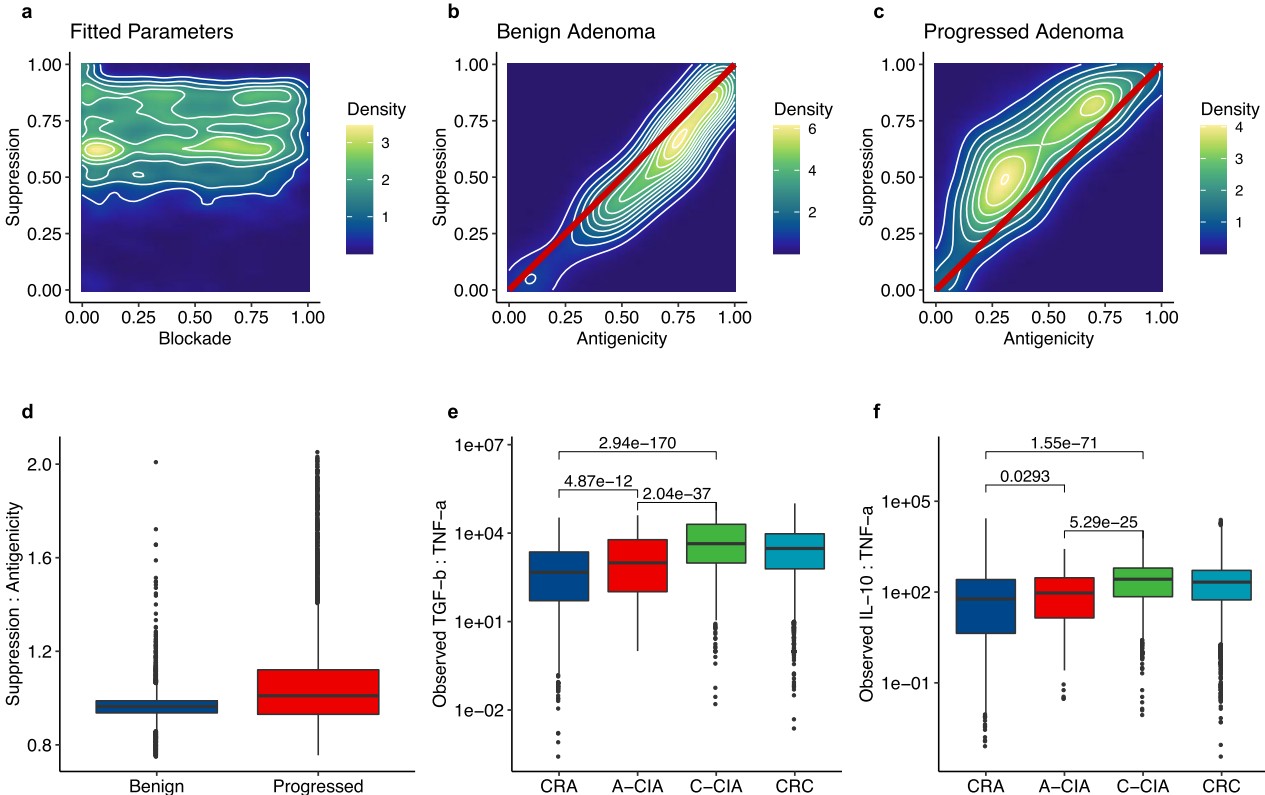

**Fig. 9 An accept/reject statistical inference method was used to determine which model parameters could recreate the observation that benign adenomas (CRA) were more immunogenic and also had more immunosuppressive cells than progressed adenomas (A-CIA).** **a** Heatmap showing the number of times each combination of Blockade ($\phi$) and Suppression ($\sigma$) produced a benign CRA that had higher antigenicity, more immune suppression, and more immune blockade than A-CIA using that same parameter combination. The narrow range of high suppression values (0.583–0.83, 25th and 75th percentiles, respectively), and wide range of immune blockade values (0.167–0.71, 25th and 75th percentiles, respectively), suggests that, in colorectal cancer, immune suppression is strong, and the effect of the blockade is minimal, as there is no clear gradient along the Blockade axis. **b**, **c** Heatmaps showing the relationship between antigenicity and immune suppression of simulated benign and progressed adenomas that fit the data, with the red line showing a 1:1 relationship. For any given antigenicity, simulated progressed adenomas have more immune suppression compared to benign adenomas. **d** Ratio of suppression to antigenicity in benign and progressed adenomas in the 8,571 simulations that fit the data. Progressed adenomas' higher ratio allows them to effectively reduce immune predation. **e**, **f** Observed spatial ratios of immunosuppressive cytokines (TGF-β, IL-10) to inflammatory cytokines (TNF-α), based on marker abundances within each samples' quadrats ($n = 9$ independent CRA, $n = 9$ independent CIA (A-CIA and C-CIA), and $n = 9$ independent colorectal carcinomas (CRC)). Dunn's test was used to calculate the unadjusted p-values. As predicted by modeling, compared to benign adenomas (CRA), progressed adenomas (A-CIA) have significantly higher ratios of immunosuppressive cytokines to inflammatory cytokines. While the effects of cytokines may not be equal, these results suggest that the effect of immune suppression will be greater in progressed adenomas than in benign adenomas. In **d**–**f**, the center line of each boxplot indicates the median, the top and bottom of the box indicate the 75th and 25th percentiles, respectively, the top whisker the largest value that is no further than 1.5 interquartile range (IQR) from the 75th percentile, the bottom whisker the smallest value no more than 1.5 IQR from the 25th percentile, and points indicate outliers.

progression of human CRC. Benign adenomas are unable to pass through an immunogenic bottleneck due to high immunogenicity and insufficient immune suppression. Conversely, adenomas that progress do so by avoiding cytotoxic T cell infiltration via immune suppression that is sufficiently strong to overcome their significantly lower immunogenicity. The immunosuppressive niche of progressed adenomas continues to expand during progression to carcinoma, resulting in a highly proliferative tumor existing within an immune-cold ecology (Fig. 10).

This conclusion is supported by the observations that, compared to progressed adenomas (A-CIA), benign adenomas (CRA) exhibit significantly higher abundances of cytotoxic T cells, likely due to their also significantly higher immunogenicity (Figs. 5d and 8a). These higher levels of cytotoxic T cells and immunogenicity are so severe that they drive unique ecological and genomic signatures for CRA (Figs. 6a and 8c). CRA also exists within a spatial community of both pro- and anti-tumor immune cells (Fig. 7c), suggesting that the immune system is successfully mounting an attack against the tumor, despite the presence of

immunosuppressive cells and cytokines. In contrast, progressed adenomas reside within a predominately immunosuppressive spatial community (niche), separated from cytotoxic T cells (Fig. 7d). While progressed adenomas (A-CIA) have similar abundances of immunosuppressive cells and cytokines, another critical difference is that A-CIA has significantly fewer cytotoxic T cells and lower antigenicity (Figs. 5d and 8a). Together, these observations suggest that, unlike benign adenomas (CRA), the combination of lower immunogenicity and an immunosuppressive niche more successfully reduces immune predation, which will allow the tumor to persist and eventually progress. Throughout progression from adenoma to carcinoma (A-CIA to C-CIA to CRC), there is a trend of decreasing abundances of cytotoxic T cells and TNF-α, coupled with increases of tumor, M2 macrophage markers, and immunosuppressive cytokines (Fig. 5d), suggesting that immune suppression continues to strengthen. The net result of these changes is the generation of a homogenous immune-cold ecology common to carcinomas, allowing them to reside within a spatial community isolated from

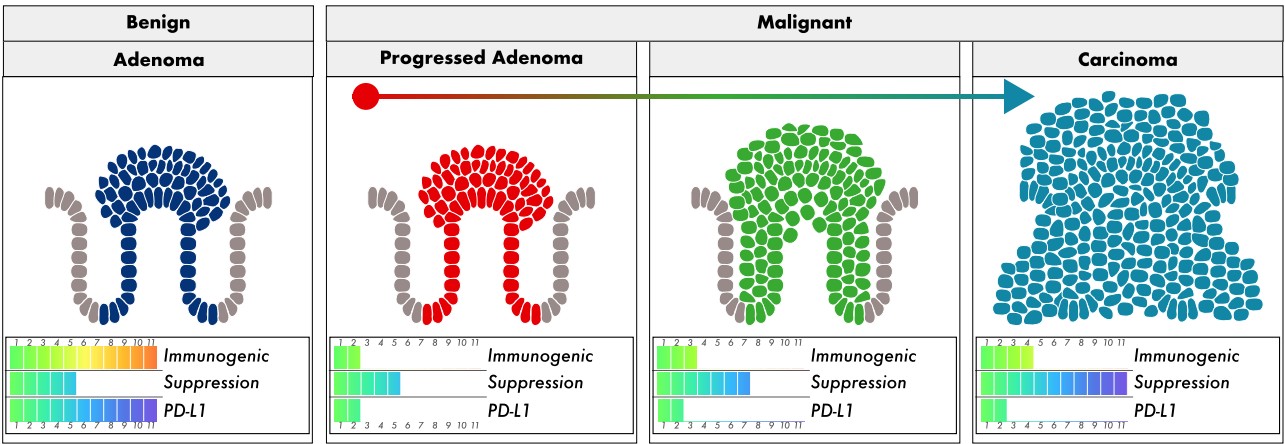

**Fig. 10 Progressed adenomas construct an immunosuppressive niche.** This suggests that benign CRA are stuck at the immunogenic bottleneck. CRA that do progress instead appear to initially survive using the Get Lucky strategy, while eventually shifting to a Get Smart one as immune suppression strengthens. As immune escape relaxes selection against antigenicity, colorectal carcinomas (CRC) rely less on Get Lucky, and are able to accumulate neoantigens, increasing neoantigen burden. These conclusions are consistent with the observation that antigenicity and cytotoxic T cells are good prognostic markers, but also suggest that altering the immunosuppressive ecology, as opposed to targeting checkpoint inhibitors, may prove to be a more successful treatment strategy.

the immune system, where they are free to divide and accumulate neoantigens (Figs. 6b, d, 7f, and 8a–c).

Our data-driven hypothesis that CRA remain controlled due to an anti-tumor immune response that is minimally suppressed is consistent with our mathematical model, where we showed that benign adenomas have a lower ratio of immune suppression to antigenicity, compared to adenomas that progressed (Fig. 9b). While it has been shown that cytotoxic T cell infiltration is a marker of good prognosis[18–20], and immune suppression is a marker of poor prognosis[12,59,60], our hypothesis suggests that it is important to consider both inflammation (e.g., cytotoxic T cells, M1 macrophages, inflammatory cytokines, etc.) and immune suppression when assessing the risk of progression. If we had considered only the abundance of immunosuppressive cells and cytokines, we might have predicted that our benign adenomas would progress. However, when we incorporated tumor immunogenicity and the spatial organization of the tumor microenvironment, we saw clear signs that the immune system was mounting a response against the tumor, despite the presence of immunosuppressive cells and cytokines. In other words, without considering both pro- *and* anti-tumor inflammation, we might have mistakenly predicted the adenoma had a high risk of progression.

A second clinical implication of our work is that immune blockade appears to play little role in the pathway to malignancy. Modeling suggests that immune suppression is the superior strategy and that it will outcompete those relying solely on blockade (Figs. 1 and 3). This is because, similar to the immune blockade, immune suppression offers protection, but also creates an environment more supportive of tumor growth. These features of immune suppression make blockade redundant: if there is already immune suppression, blockade offers little to no additional benefit. Our ecological analysis reinforces this prediction that immune suppression obviates the need for the immune blockade, for example when compared to benign adenomas, PD-L1 is found at significantly lower levels in progressed adenomas and carcinomas, suggesting it does not play an important role in tumorigenesis or progression (Fig. 5d). Thus, we suspect that while PD-L1 may help prevent CRA from being eliminated at the very earliest stages, it is not an important feature of any stage of the pathway from A-CIA to C-CIA to CRC.

Our hypothesis that PD-L1 does not play a large role in tumorigenesis or progression may help explain why MSI-low tumors respond poorly to immune checkpoint inhibitors[61–63]. Similarly, given that immune suppression appears to be the dominant escape strategy, and that CD8 T cells are isolated from the tumor, we predict that therapies designed to increase cytotoxic T cell killing, such as dendritic cell vaccines and chimeric antigen receptor (CAR) T cell therapy, would have limited success unless the immunosuppressive niche is addressed with additional therapy.

Given that the immunosuppressive niche seems to be the driver of immune escape, more effective treatments of CRC might be those that seek to re-engineer a hot immune ecology, possibly by re-polarizing immunosuppressive cells, a treatment currently being explored for macrophages[64–67]. An additional benefit to re-engineering the immunosuppressive niche is that doing so could shift the fitness landscape for all cells within the tumor. This is in contrast to targeted therapies aimed at eliminating only those cells that carry a particular mutation, an approach that suffers from almost inevitable acquired resistance and relapse. Thus, a major benefit to re-engineering the immunosuppressive niche is that by targeting a common underlying mechanism of tumorigenesis, it casts a wider net on the heterogeneous population of cells, potentially reducing the risk of evolving resistance.

In summary, we provide evidence of a critical role for immune predation in preventing colorectal malignancy, implying that immune evasion represents a key bottleneck in disease progression. In CRC, our analysis strongly suggests that it is the construction of an immunosuppressive niche by the tumor that is the predominant pathway through this bottleneck. As the immunosuppressive niche is fundamental to progression and persistence, re-engineering the microenvironment towards an immune-hot phenotype may prove to be an effective form of immunotherapy.

## Methods

**Sample collection and processing.** FFPE samples ($n = 54$) representing adenomas (CRA, $n = 13$), adenomas with foci of cancer ("ca-in-ads", CIA, $n = 24$), and carcinomas (CRC, $n = 17$) were selected from the histopathology archives of University College Hospital, London, under UK ethical approval (07/Q1604/17) or John Radcliffe Hospital, Oxford under ethical approval (10/H0604/72). Written informed consent was waived by the relevant RECs due to the retrospective and anonymous nature of this study. From each block, 7 serial sections were taken at

4-micron thickness, the first was stained with hematoxylin and eosin (H&E) and used for histopathological classification by two expert pathologists (M.R.-J. and M.J.). The remaining 6 sections were used for dual-color IHC staining. A further 6 sections at 5 micron were taken from a subset of blocks ($n = 10$) and used for DNA extraction.

**DNA extraction**. Different histopathological regions within individual lesions were demarcated on H&E slides by an experienced GI pathologist. This was used to guide careful needle dissection and DNA was then extracted from these discrete areas using the DNA QIAamp Mini Kit (Qiagen) and standard protocols.

**Whole-exome sequencing**. Multi-region WES was performed on a subset of samples representing CRA ($n = 4$ with two regions each), CIAs ($n = 3$, with one region from the carcinoma region and two from the adenoma region), and CRC ($n = 3$, with two regions each). The quality of extracted FFPE DNA was verified using a multiplex PCR as previously described (van Beers et al.[68]). Briefly, 1 ng of FFPE DNA is used as the template for a multiplex PCR reaction using four sets of primers against the GAPDH gene, generating products of 100, 200, 300, and 400 bp in length. Amplification of high-quality FFPE DNA produces all four amplicons; however, DNA that is heavily degraded will only produce the shorter fragments. Only DNA samples that showed successful amplification of fragments >300 bp in length were considered for WES. DNA input of 50 ng was used to prepare sequencing libraries with the Nextera Rapid Capture Exome kit, according to the manufacturer's instructions (Illumina, Cambridge, UK). Libraries were sequenced with a target depth of 60× on Illumina's HiSeq 2500 with 125 bp paired-end reads (v4 chemistry). Additional samples were added from (Sottoriva et al.[69]) adding to the total number of multi-region CRA and CRC ($n = 3$ and $n = 3$, respectively). Alignments to the hg19 reference genome were conducted using the BWA-mem algorithm[70] and processed using the GATK best practices workflow (Van der Auwera et al.[71]) for downstream analysis.

**Variant calling**. Prior to calling variants, we normalized binary alignment/map (BAM) by downsampling to the lowest observed average depth across samples. First, we calculated the average depth for target capture regions using samtools 1.2[72]. Once the average depth was calculated for each sample, the proportion of reads needed for each sample to reach the minimum average depth was calculated. Down-sampling was then performed using this proportion for each BAM file using PICARD 2.17 (Broad Institute, 2017). This was conducted on each sample's region to generate ten replicate BAM files. Variant calling was then performed for each replicate sample group (multiple regions against normal) using multiSNV 2[73], a joint calling method specifically designed for multi-region same patient experimental designs. Criteria used for assessing variant candidates dictated a minimum mapping quality of 30, a minimum base quality of 20, with at least five reads in the tumor and normal regions, and two variant alleles. Once a variant call set was obtained variants were further scrutinized for additional criteria. A total of ten reads were required within all normal sites and variant sites with no variant alleles present in the normal. Furthermore, a variant must be supported by a minimum number of two variant reads for at least one region; while the minimum variant allele frequency in one region is 0.1.

**Neoantigen predictions**. Human leukocyte antigen (HLA) haplotypes (A, B, and C) were called using PolySolver (Shukla et al.[74]) prior to downsampling on all normal regions for each patient. Neoantigen predictions were performed using NeoPredPipe 1.0 (Schenck, Lakatos, Gatenbee, Graham, & Anderson, 2019), which utilizes ANNOVAR[75] for variant annotations and NetMHCpan 4.0 (Jurtz et al.[76]), NeoPredPipe is specifically designed to handle multi-region sequence samples. Only MHC-class I neoantigens were assessed for peptides of 8, 9, and 10-kmer lengths. A minimum cut-off of 500 nM binding affinities were used to be considered a putative neoantigen. To assess T cell receptor binding potential, Łuksza et al.'s[77] recognition potential algorithm implemented within NeoPredPipe was used.

**Dual-color IHC**. Sequential dual-color IHC of 10 markers was performed according to standard protocol. Briefly, 4 μm serial sections were dewaxed, rehydrated, and immersed in 3% hydrogen peroxide for 20 min to quench endogenous peroxidase activity. Antigen retrieval was carried out at 95 °C for 20 min in sodium citrate buffer (pH 6.0) unless otherwise specified (Supplementary Table 2). After cooling, sections were incubated with blocking buffer (phosphate-buffered saline supplemented with 2% goat serum and 1% bovine serum albumin) for 1 h at RT. Primary antibodies were diluted in blocking buffer and applied for 1 h at RT or overnight at 4 °C (see Supplementary Table 2 for antibody details). Sections were then incubated with a biotinylated secondary antibody at RT for 45 min, followed by incubation with streptavidin–biotin peroxidase solution at RT for 45 min. Visualization of the first antibody binding was carried out using DAB, according to the manufacturer's instructions (Vector Labs, Peterborough, UK). Slides then underwent a second round of antigen retrieval, generally at 95 °C for 5 min in sodium citrate buffer (pH 6.0), before applying the blocking buffer for a further 1 h at RT. The second primary antibody was then applied (see Supplementary Table 2 for details), followed by incubation with a biotinylated secondary antibody at RT

for 45 min and incubation with streptavidin–alkaline phosphatase. Visualization of the second antibody binding was performed using Fast Red, according to the manufacturer's instructions (Abcam, Cambridge, UK). Finally, sections were lightly counterstained using Gill's hematoxylin and allowed to dry before mounting and digitizing using the Pannoramic 250 high-throughput scanner (3D Histech, Budapest, Hungary).

**In situ hybridization**. Dual-color RNA ISH was performed to detect the expression of TGFβ, TNFα, IL6, and IL10 using commercially available reagents (Advanced Cell Diagnostics, Newark, CA). The RNAscope 2.5 HD Duplex Reagent Kit (catalog number 322430) was used according to the manufacturer's instructions with the probes Hs-TGFB1 (400881), Hs-TNFA-C2 (310421-C2), Hs-IL10 (602051), and Hs-IL6-C2 (310371-C2).

**Reporting summary**. Further information on research design is available in the Nature Research Reporting Summary linked to this article.

## Data availability

The quadrat count data gathered for each image and the multi-region neoantigen predictions are available on Zenodo.

## Code availability

Code used to conduct simulations is available on GitHub.

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

## Acknowledgements

C.G., M.R.-T., and A.R.A.A. are supported by Physical Sciences Oncology Network (PSON) grant from the National Cancer Institute (grant no. U54CA193489). M.R.-T., J.W., and A.R.A.A. are also supported by the Cancer Systems Biology Consortium grant from the National Cancer Institute (grant no. U01CA23238). A.R.A.A. would also like to acknowledge support from the Moffitt Cancer Center of Excellence for Evolutionary Therapy. T.G. is supported by the Wellcome Trust (202778/Z/16/Z) and Cancer Research UK (A19771). A.S. is supported by the Wellcome Trust (202778/B/16/Z) and Cancer Research UK (A22909). We acknowledge funding from the National Institute of Health (NCI U54 CA217376) to A.S. and T.A.G. This work was also supported by a Wellcome Trust award to the Centre for Evolution and Cancer at the Institute of Cancer Research (105104/Z/14/Z). T.A.G. and S.L. are grateful for support from the Bowel and Cancer Research Charity (project grant scheme). T.A.G. received support from the Barts Charity (pilot grant scheme). S.L. was supported by Wellcome Trust Senior Clinical Research Fellowship (206314/Z/17/Z).

## Author contributions

C.D.G. developed and analyzed the computational model, developed the methods to segment and align the immunohistochemistry, and performed the analysis of the data collected from immunohistochemistry and neoantigen predictions. A.-M.B. designed, optimized, and performed all laboratory experiments, with the assistance of M.P.N. and S.Y.H. R.O.S. processed whole-exome sequencing of samples and downstream data generation and W.C.H.C., P.M., E.L., and A.S. provided additional support in genomic analysis. M.R.-J., S.L., and A.S. graciously provided samples and/or data. M.R.-J. and M.J. provided the histopathological assessment. C.J.W. provided valuable feedback on the ecological analysis. M.R.-T., T.A.G., and A.R.A.A. guided the research and supervised the writing of the manuscript. T.A.G. and A.R.A.A. conceived and funded the study.

## Competing interests

The authors declare no competing interests.
