## [Peer Review File · Nature Communications]

Reviewers' Comments:

Reviewer #1:

Remarks to the Author:

The authors have addressed my comments in a satisfactory matter, and I have no further concerns.

Reviewer #2:

Remarks to the Author:

In general, the authors have included new analyses that strengthen the manuscript, and I am supportive of its publication pending minor revisions.

Point 1 – One remaining concern is the imprecise use of the term 'inflammation' in the paper, which the authors seemingly use as synonymous with T cell responses. While 'inflammation' is something of a catch-all term and is applied in many contexts, the process of inflammation often refers to the secretion of inflammatory cytokines (IL-1, IL-6, TNF α) and other acute inflammatory mediators, such as prostaglandins, complement, and vasoactive molecules, which collectively promote activation and recruitment of immune cells to a site of damage or infection. The authors should be more precise in their terminology and discuss T cell responses when the data are, in fact, focused on T cells.

Point 2 - The authors misinterpreted the concern in the analysis of ecological heterogeneity. As such, it is still not clear why C-CIA, which ultimately represents tumor that is already invasive, and therefore must have already achieved immunosuppression according to the model, should be more variable as a class compared to the class CRAs, only some of which would have gone on to progress if they had not been surgically resected. If CRAs are uniformly characterized by stronger T cell responses, one would imagine that the subset of CRAs that would have progressed to CRC should also exhibit greater early signs of immunosuppression, as predicted by the model. This should result in a higher degree of variability within the class of CRAs since only a subset should achieve this immunosuppressive state that facilitates the future transition to malignancy. Therefore, the result still seems contradictory to the prediction that immunosuppression is an early feature of benign lesions that enables progression to malignancy. In other words, if immunosuppression precedes transformation, why are adenomas not variable with respect to their level of immunosuppression, resulting in higher ecological heterogeneity between them? This should be addressed by the authors.

Perhaps the subsequent spatial analysis hints at an explanation, that the localization of cells is actually more important than just their abundance, but the initial result is still counter intuitive. Moreover, the model does not account for differential cellular localization, though perhaps it contributes to the efficiency of suppression and is therefore captured to some extent by the 'suppression' term.

While not the intended question, the analysis of ITH is interesting.

I do not have any remaining comments about the other issues previously discussed.

Reviewer #1 (Remarks to the Author):

The authors have addressed my comments in a satisfactory matter, and I have no further concerns.

We are glad we were able to address the Reviewer's concerns, and would like to thank them for their constructive feedback.

Reviewer #2 (Remarks to the Author):

In general, the authors have included new analyses that strengthen the manuscript, and I am supportive of its publication pending minor revisions.

Point 1 – One remaining concern is the imprecise use of the term ‘inflammation’ in the paper, which the authors seemingly use as synonymous with T cell responses. While ‘inflammation’ is something of a catch-all term and is applied in many contexts, the process of inflammation often refers to the secretion of inflammatory cytokines (IL-1, IL-6, TNF α) and other acute inflammatory mediators, such as prostaglandins, complement, and vasoactive molecules, which collectively promote activation and recruitment of immune cells to a site of damage or infection. The authors should be more precise in their terminology and discuss T cell responses when the data are, in fact, focused on T cells.

The Reviewer makes a good point, and we have gone back through the manuscript to clarify our use of inflammation, as either referring to the collection of anti-tumor immune cells (M1 macrophages, cytotoxic T cells) and associated cytokines (TNF- α , IL-6), or specifically cytotoxic T cells.

Point 2 - The authors misinterpreted the concern in the analysis of ecological heterogeneity. As such, it is still not clear why C-CIA, which ultimately represents tumor that is already invasive, and therefore must have already achieved immunosuppression according to the model, should be more variable as a class compared to the class CRAs, only some of which would have gone on to progress if they had not been surgically resected. If CRAs are uniformly characterized by stronger T cell responses, one would imagine that the subset of CRAs that would have progressed to CRC should also exhibit greater early signs of immunosuppression, as predicted by the model. This should result in a higher degree of variability within the class of CRAs since only a subset should achieve this immunosuppressive state that facilitates the future transition to malignancy. Therefore, the result still seems contradictory to the prediction that immunosuppression is an early feature of benign lesions that

enables progression to malignancy. In other words, if immunosuppression precedes transformation, why are adenomas not variable with respect to their level of immunosuppression, resulting in higher ecological heterogeneity between them? This should be addressed by the authors.

Perhaps the subsequent spatial analysis hints at an explanation, that the localization of cells is actually more important than just their abundance, but the initial result is still counter intuitive. Moreover, the model does not account for differential cellular localization, though perhaps it contributes to the efficiency of suppression and is therefore captured to some extent by the 'suppression' term.

We now better understand the Reviewer's concern regarding the observed lower intra-stage heterogeneity (ISH) of CRA, as compared to that of C-CIA. We believe the lower ISH of CRA could occur for several reasons:

1. As the Reviewer suggests, while CRA share a similar mix of cells (resulting in low intra-stage heterogeneity), it very well could be that the more important factor is the spatial configuration cells within the tumor.
2. Epidemiological data and direct observation shows that progression from adenoma to carcinoma is very rare (Logan et al., 2012; Zauber et al., 2012; Hofstad et al., 1996). Thus, within our cohort of CRAs, it is likely very few, if any, of the CRAs will be on a trajectory to progress to cancer in the near to medium term. Consequently, any signal from an "outlier" progressor-CRA (i.e. those CRA that would have progressed had they not been resected) would be low, meaning that we should actually expect *most* CRAs to have similar (non-progressor) ecologies, resulting in low ISH.
3. Modeling predicts that the transition to CRC is rapid when immune suppression is strong (Figure 3C), and so catching a CRA at the very earliest stages of transition (i.e. "catching it in the act of transforming") could be rare. As is the case with the second hypothesis, the signal from such rare CRA would be drowned out by the majority of CRA that remain under immune control.
4. A-CIA and C-CIA may have higher ISH than CRA due to their being resected at different points in the transition to an immune cold ecology. That is, unlike CRA and CRC, which may have a stable ecology, the CIA samples (A-CIA and C-CIA) are in a state of transition from CRA to CRC, having been resected at different points along this trajectory (and their may also be further spatial heterogeneity as the reviewer points out). Some may be in the earlier stages, some at the late stages, and some in between. Given that many microenvironmental changes would occur during this transition, and that we are essentially sampling that transition at different time points, it could be expected that the resected A-CIA and C-CIA exhibit a wide variety of ecologies, resulting in the observed higher ISH.

As suggested by the Reviewer, we have expanded our discussion of ISH to more fully articulate the reasons behind the observed patterns, using the above hypotheses.

While not the intended question, the analysis of ITH is interesting.

Even though the analysis of ITH did not actually address the Reviewer's concern, we too were happy to see that it was interesting and contributed to the understanding of the rest of the work.

I do not have any remaining comments about the other issues previously discussed.

We are happy to hear the Reviewer was satisfied with most of our updates. We believe addressing their concerns, particularly those related to the spatial analysis, has greatly improved the paper.

Reviewers' Comments:

Reviewer #2:

Remarks to the Author:

I am happy to see that the reviewers have responded to the remaining comments and incorporate the discussion into the manuscript. I have no further comments at this time.

REVIEWERS' COMMENTS

Reviewer #2 (Remarks to the Author):

I am happy to see that the reviewers have responded to the remaining comments and incorporate the discussion into the manuscript. I have no further comments at this time.

We are delighted to hear that the reviewer is satisfied with our efforts to address their remaining comments.